# Synergistic Interaction Between Endophytic *Bacillus pumilus* and Indigenous Arbuscular Mycorrhizal Fungi Complex Improves Photosynthetic Activity, Growth, and Yield of *Pisum sativum*

**DOI:** 10.3390/plants14131991

**Published:** 2025-06-30

**Authors:** Mounia Akhallaa Youne, Oumnia Akhallaa Youne, Mohammed Bouskout, Yaseen Khan, Hamza Khassali, Sulaiman Shah, Ahmed Sujat, Hassan Alahoui, Mohamed Najib Alfeddy, Bacem Mnasri, Lahcen Ouahmane

**Affiliations:** 1Cadi Ayyad University, Faculty of Sciences-Semlalia, Laboratory of Water Sciences, Microbial Biotechnologies, and Natural Resources Sustainability, Unit of Microbial Biotechnologies, Agrosciences, and Environment-CNRST Labeled Research Unit N°4, P.O. Box 2390, Marrakesh 40000, Morocco; m.akhallaayoune.ced@uca.ac.ma (M.A.Y.); o.akhallaayoune.ced@uca.ac.ma (O.A.Y.); mohammed.bouskout@ced.uca.ma (M.B.); hamza.khassali@ird.fr (H.K.); 2Center for Eco-Environment Restoration Engineering of Hainan Province, School of Ecology, Hainan University, Haikou 570228, China; 3School of Life Sciences, Northeast Normal University, Changchun 130024, China; sul209@nenu.edu.cn; 4School of Agriculture and Environment, Institute of Agriculture, The University of Western Australia, Perth, WA 6001, Australia; sujat.ahmed@research.uwa.edu.au; 5Fellah Palm, Limited Liability Company (LLC), Marrakesh 40000, Morocco; hassan.alahoui@fellahpalm.com; 6Phytobacteriology Laboratory, Plant Protection Research Unit, CRRA, National Institute for Agronomical Research, Marrakesh 40000, Morocco; mohamednajib.alfeddy@inra.ma; 7Laboratory of Legumes and Sustainable Agroecosystems, Centre of Biotechnology of Borj-Cédria, BP 901, Hammam-Lif 2050, Tunisia; mnbacemm@yahoo.com

**Keywords:** *Bacillus pumilus*, chlorophyll fluorescence, mycorrhizal fungi complex, *Pisum sativum*, plant growth-promoting rhizobacteria, SPAD index

## Abstract

The demand for sustainable agriculture has prompted the exploration of alternative methods to boost crop growth and yield. Microbial biostimulants offer effective solutions to enhance plant performance and reduce reliance on chemical fertilizers. This study investigated the effects of *Bacillus pumelo* (*B. pumilus*), applied individually and in combination with a mycorrhizal fungi complex, on the growth, yield, and photosynthetic activity of pea (*Pisum sativum*). Pea seeds were grown in sterilized soil under four treatment conditions, including a non-inoculated control, inoculation with 2.5 mL of *B. pumilus* culture per seedling, inoculation with an indigenous mycorrhizal fungal complex, and a combined treatment of *B. pumilus* and the mycorrhizal complex. The biostimulant treatments significantly influenced all measured photosynthetic and growth parameters. The results showed that *B. pumilus* substantially promoted pea growth, leading to notable improvements in biomass, plant height, and photosynthetic efficiency. When combined with the mycorrhizal fungi complex, these growth-promoting effects were significantly amplified, resulting in a ~69.7% increase in shoot fresh weight, a ~72.7% rise in root dry weight, and a ~73.6% boost in flower production. Additionally, the chlorophyll content increased by ~180% and photosynthetic yield (F*v*/F*m*) improved by ~18.5%. The combined treatment also produced the highest SPAD index value, reflecting a ~57% increase. The synergistic interaction between *B. pumilus* and mycorrhizal fungi enhances photosynthetic efficiency and overall plant performance. The study highlights the potential of using these microbial inoculants as biostimulants to improve pea cultivation in agroecosystems, offering a sustainable alternative to chemical fertilizers.

## 1. Introduction

Agriculture has long been the cornerstone of human civilization, fueling the industrial revolution and laying the foundation for the secondary and tertiary sectors of the economy. As the primary sector, agriculture drives economic growth and provides stable employment, particularly for vulnerable populations [1]. A resilient agricultural sector is integral to poverty alleviation, economic transformation, and ensuring food security, especially in the face of global challenges such as climate change [2]. Unfortunately, agriculture faces mounting pressures from extreme weather events, rising soil salinity, and inadequate subsidies [3]. These challenges exacerbate the global food crisis, undermine agricultural productivity, and threaten food security worldwide.

Legumes, particularly peas (*Pisum sativum*), serve as a vital nutritional source, especially during the winter months, providing protein, calories, and essential minerals. Beyond their nutritional importance, legumes play a critical role in soil health, enhancing soil fertility by fixing nitrogen and enriching the soil with other nutrients, such as phosphorus, through symbiotic relationships with soil bacteria. Despite these benefits, legume cultivation in Morocco remains stagnant, with considerable fluctuations in yield. Between 1961 and 2020, legumes were grown on an average of 400,000 hectares, representing only ~3.7% to ~8% of Morocco’s total agricultural area of around 8.7 million hectares. By 2021, this figure had fallen to just ~300,000 hectares [4]. The total area devoted to legume cultivation reaches around 270,000 ha, representing ~4% of the UAA in 2022 [5]. During favorable agricultural seasons, this area can produce up to 2 million quintals of crops. Among the various legumes grown, beans are in first place in terms of their contribution to national production, accounting for ~35% of the total, followed by chickpeas, which account for ~25% of this production. Lentils are in third place with a share of ~17%, while peas account for 13%. The other pulses grown in the country account for the remaining ~10% [5].

Domestic production covers only ~24% of the country’s legume consumption needs [6], forcing Morocco to increasingly rely on imports, which surged from ~198,000 quintals in 2009 to ~538,000 quintals in 2019. Peas, which represent ~13% of the total legume-growing area in Morocco, rank fourth in importance after beans, chickpeas, and lentils. However, the profitability of pea cultivation remains low compared to other cereal crops due to challenges such as drought, diseases, pests, and weeds. Broomrape infestations remain a significant issue, hampering crop development and rendering affected fields unsuitable for cultivation [6].

In response to these challenges, biostimulants, such as biofertilizers, are being used as sustainable solutions to improve soil fertility and enhance plant growth, as well as mitigate environmental stressors [7]. These biostimulants often contain plant growth-promoting bacteria (PGPB) and arbuscular mycorrhizal fungi (AMF), both of which have been shown to improve plant performance by promoting nutrient uptake, enhancing resistance to biotic and abiotic stresses, and boosting overall growth [8,9]. Recent research has shown the promising impact of microbial biostimulants on *Pisum sativum* growth, particularly in semi-arid environments. For instance, a two-year field study in Tunisia demonstrated that inoculations with PGPB, including *Rhizobium laguerreae* and *Erwinia* sp., improved productivity and biometric parameters. The study also highlighted significant increases in soil bacterial and AMF diversity, indicating the synergistic roles of microbial consortia in enhancing soil health and pea growth [10]. PGPBs such as *Bacillus pumilus* (*B. pumilus)* are particularly promising in this regard. *B. pumilus* has been shown to enhance plant growth through mechanisms like auxin production, nitrogen fixation, and phosphorus solubilization [11]. Furthermore, it produces antimicrobial compounds that help protect plants from pathogens [12,13,14]. Notably, *B. pumilus* has also demonstrated the potential to improve plant stress tolerance, particularly under drought conditions [15]. Additionally, *B. pumilus* has been shown to facilitate the bioremediation of organic pollutants and enhance soil microbial diversity, further promoting sustainable agriculture [16].

Likewise, arbuscular mycorrhizal fungi (AMFs) are essential for sustainable agriculture because they can form mutualistic relationships with plant roots. AMFs enhance plant nutrient uptake, particularly phosphorus and nitrogen, and increase the plant’s resilience to various environmental stresses [17,18,19]. Moreover, AMFs contribute to soil health by increasing the microbial biomass, improving soil structure, and promoting nutrient cycling [20]. The synergistic effects of AMFs and PGPBs have garnered significant interest, as these microorganisms can work together to enhance plant growth and stress tolerance, further improving crop yield and quality [21,22,23].

This study investigates the combined effects of a *B. pumilus* strain isolated from the rhizosphere of *Acacia cyanophylla* and an autochthonous mycorrhizal fungal complex on the growth, yield, and photosynthetic performance of *Pisum sativum* under greenhouse conditions. This research aims to evaluate the potential synergistic interactions between these microbial inoculants and assess their abilities to improve pea production in Morocco. By exploring this interaction, the study aims to provide a sustainable alternative to conventional farming practices, thereby enhancing legume cultivation in Morocco’s Mediterranean agroecosystems, improving food security, and contributing to sustainable agricultural practices in a changing climatic context.

## 2. Materials and Methods

### 2.1. Mycorrhizal Inoculum Production

The inoculum consisted of a mixture of indigenous AMFs present in rhizosphere soil samples (which contained ~1800 spores per 100 g) and an aliquot of infected root fragments from trapped maize plants. The choice of maize (*Zea mays* L.) for this purpose was based on its high germination rate, early sensitivity to mycorrhizal colonization, and abundant root growth. Additionally, trap culture has been utilized to enhance the naturally associated native mycorrhizal complex of *Argania spinosa* in the Essaouira region. The trap culture technique was used following the method outlined by Ouahmane et al. [24]. Initially, the seeds of maize were sterilized on the surface with a 1% sodium hypochlorite (NaOCl) solution for 15 min, washed with distilled water multiple times, and germinated at a temperature of 25 °C (for 48 h) before sowing. Five pregerminated maize seeds (rootlets ~1–2 cm long) were transplanted into each 1 kg pot containing a mix (1*v*/1*v*) of sterilized rhizospheric soils/sandy soil (121 °C for one hour on three consecutive days) for four months under greenhouse conditions.

### 2.2. AM Fungal Identification and Diversity

The investigation focused on the molecular identification of the mycorrhizal complex. Fungal DNA was extracted from a subsample of 40 mg of surface-sterilized *Zea mays* roots cultivated in *Argania spinosa* rhizosphere soil (ground using a FastPrep-24 homogenizer (MPbiomedicals Europe, Illkirch, France)). DNA was extracted using a Fast DNA VR SPIN kit (MP Biomedicals Europe, Schwege, Germany) according to the manufacturer’s instructions. Guanidine thiocyanate was added to improve the extraction process. DNA extracts were purified by adding 20–30 mg of polyvinylpolypyrrolidone (PVPP) to limit the presence of polymerase chain reaction (PCR) inhibitors. Fungal DNA amplification targeted the 18S rRNA gene that was amplified using the primers NS31 and AML2 [25,26]. The resulting PCR products were purified, spiked with PhiX (15%), and loaded onto the Illumina MiSeq cartridge (Illumina, San Diego, CA, USA) according to the manufacturer’s instructions. The molecular identification of AM fungi was performed using Illumina MiSeq sequencing of the 18S rRNA gene region. Raw reads were processed and analyzed using the bioinformatic pipeline detailed by Thioye et al. [27], including primer clipping, quality filtering, chimera removal, and taxonomic assignment using the Stampa pipeline and the PR2 database.

### 2.3. Bacterial Inoculum Production

The *Bacillus* strain was isolated from the rhizosphere of *Acacia cyanophylla* in the Essaouira region, located on a sand dune within a semi-arid climate. This environment is characterized by temperature fluctuations between 18 °C and 22 °C, along with low annual precipitation averaging approximately 295 mm. The bacterial inoculum was prepared based on a strain of *Bacillus* obtained by culturing the bacteria in solid nutrient media (yeast extract–mannitol medium), and then inoculated in liquid YEM medium with stirring for 72 h at 25 °C [28].

### 2.4. Bacterial Identification

The bacteria were characterized based on their nearly complete 16S rDNA sequences and partial 16S-23S rRNA ITS. A loopful of bacterial cells, obtained from the margin of the bacterial colony, was suspended in 20 µL of sterile water, and cellular debris was removed by centrifugation at 13,000 rpm for 1 min at room temperature. Subsequently, 2 µL of the supernatant was utilized as a template for PCR. The partial ITS of the 16S and 23S rRNA genes was amplified using the primers BR5 (5′-CTTGTAGCTCAGTTGGTTAG-3′; [29]) and FGPL132′ (5′-CCGGGTTTCCCCATTCGG-3′; [30]). Each PCR amplification was carried out in a 25 µL reaction tube containing 2 µL of bacterial DNA template, 1× Green GoTaq^®^ Reaction Buffer (1.5 mM MgCl_2_), 200 µM of each dNTP, 0.4 µM of each primer, and 1.25 U of GoTaq^®^ DNA polymerase (Promega, Tokyo, Japan), with the following temperature cycles: an initial cycle of denaturation at 95 °C for 5 min; 35 cycles of denaturation at 95 °C for 30 s, annealing at 55 °C for 30 s, and extension at 72 °C for 40 s; and a final extension at 72 °C for 5 min. The PCR products were directly sequenced using the same primers as for amplification. Genoscreen performed the sequencing in Lille, France. For each sample, forward and reverse sequencing reactions were performed, and the sequences were verified through a BLAST search using the NCBI GenBank database via an online website (https://blast.ncbi.nlm.nih.gov/Blast.cgi (online version, accessed on 14 March 2025)). Later, the obtained DNA sequences were edited both manually and using BioEdit 7.0.4.1 [31] and Finch TV programs (https://finchtv.software.informer.com/1.4/ (accessed on 10 March 2025)), and visualized with the online website iTOL (interactive tree of life).

### 2.5. Seed Treatment

The pea seeds were immersed in a 3% NaOCl solution for 10 min, followed by three rinses with double-distilled water to sterilize them. After sterilization, the seeds were soaked in water for 6 h. Afterward, the seeds were placed on sterilized and moist filter paper in 9 cm-diameter disposable Petri dishes (20 seeds per dish). They were incubated in the dark in an incubator at 24 ± 1 °C for germination until the radicles emerged.

### 2.6. Experimental Design

Following the rooting support period, seedlings were transplanted into experimental pots filled with 1 kg of substrate. The pots were disinfected to prevent contamination before use. The experimental design included non-inoculated pots (control), plants inoculated using a mix of indigenous arbuscular mycorrhizal fungi (AMFs), plants inoculated with *Bacillus pumilus* (*B. pumilus)* strain, and plants inoculated with both *B. pumilus* and the AMF complex. The mycorrhizal inoculum, consisting of 2 g of infected root fragments, was placed near the seedling roots to promote fungal infection. Non-inoculated plants received the same sterilized root fragments. The bacterial inoculum comprised a 2.5 mL bacterial suspension, approximately 2.10^8^ UFC/mL, at the exponential growth stage for each pea seedling after the emergence of the main root. The plants were randomly arranged in a complete block design with 15 replicates for each treatment. Daily watering with potable water was provided as needed. The cultivation spanned three months in a greenhouse under natural daylight conditions. Temperatures during the experiment ranged from ~26 °C to ~28 °C, with a relative humidity of ~60–80%. The average daylight duration was approximately ~10 h.

### 2.7. Morphological Features and Biomass Production

Three months after planting, various morphological attributes and biomass production were estimated. The lengths of the main shoot and root were evaluated. The numbers of leaves, flowers, and pods were counted. To assess the impact of mycorrhization on the growth of pea plants, four randomly chosen plants from each treatment were uprooted from the containers, and their root parts were gently rinsed. The fresh weight (FW) of each sample was immediately calculated after harvesting using an electronic precision balance (0.001 g). Subsequently, the plant shoots and roots were oven-dried at 110 °C for 24 h to obtain their dry weights (DW).

### 2.8. Photosynthetic Activity and Yield

Chlorophyll fluorescence was determined using a chlorophyll meter. The device used was SPAD-502Plus (Ref. SPAD502PLUS). The contents of chlorophyll a and b were determined by Arnon’s method [32]. The method consisted of weighing a 1 g sample of leaves and grinding it in the presence of 25 mL of 80% acetone. The extract was filtered and then placed in the dark to avoid the oxidation of the chlorophyll by light. The dosage was determined by reading 3 mL of the solution on a spectrophotometer. The optical density was determined at 663 nm for chlorophyll a and at 645 nm for chlorophyll b. Lichtenthaler’s formulas were utilized to calculate pigment concentrations in the samples [33]. The final values were presented in milligrams of pigment per gram of fresh leaf weight (mg g^−1^ FW).

### 2.9. Mycorrhizal Colonization Rate of Pea Roots

Fresh root samples taken from the mycorrhizal groups of four individual plants were cleared and stained according to Phillips and Hayman’s method [34]. Selected root sections were cleared with 10% (*w*/*v*) potassium hydroxide (KOH) at 90 °C for one hour to remove their cytoplasmic and nuclear contents, then immersed in hydrochloric acid HCl (5%) for 5 min to neutralize the alkalinity due to KOH. Next, they were stained with 0.05% (*w*/*v*) trypan blue in lactoglycerol (1*v*/1*v*/1*v* of distilled water, glycerol, and lactic acid) for 30 min at 90 °C. The stained roots were cut into approximately 1 cm-long fragments, arranged parallel to each other on a microscopic slide in five replicates of ten root fragments each, and then carefully crushed with the coverslip. A random sample of non-inoculated plant roots was tested to check for fungal infections. The preparations were evaluated under a microscope at 40x magnification, and the mycorrhizal colonization in plant roots was estimated according to the method described by Trouvelot et al. [35]. This assessment includes the degree of root colonization, mycorrhizal frequency (F%), as well as mycorrhizal intensity (M%).

### 2.10. Statistical Analysis

To evaluate the synergistic effects of *Bacillus pumilus* and the mycorrhizal fungal mixture on the various parameters of *Pisum sativum*, we calculated the mean values ± standard deviations (SDs) for each parameter. The results were presented in both tabular and graphical formats. One-way ANOVA was performed to assess the significance of the effects of the microbial treatments on the plant variables. Before conducting the ANOVA, the data were tested for normality and homogeneity of variances to ensure they met the assumptions required for parametric testing. A *p*-value of less than 0.05 defined statistical significance. For the post hoc analysis, Tukey’s HSD test was applied to perform multiple pairwise comparisons and to group the treatments into homogeneous subsets. Principal component analysis (PCA) was performed in SPSS 25.0 to identify the first two components, and a biplot was generated to visualize the relationships between variables. Pearson’s correlation coefficients were calculated to assess the correlations among parameters, with significance set at *p* < 0.05.

## 3. Results

### 3.1. Composition and Diversity of the AMF Complex Associated with Zea mays in the Argania Spinosa Rhizosphere

The molecular analysis revealed that the arbuscular mycorrhizal fungal (AMF) complex associated with *Zea mays* in the rhizosphere of *Argania spinosa* predominantly consists of members from the phylum *Glomeromycota* and the class *Glomeromycetes*. The majority of the AMF community belongs to the order *Glomerales* (~99.07%), with the families *Glomeraceae* (~98.64%) and *Claroideoglomeraceae* (~0.43%) being the primary contributors. A smaller proportion (~0.25%) was classified under the order *Paraglomerales*, specifically within the family *Paraglomeraceae*. At the genus level, the AMF complex is primarily composed of *Glomus* (~83.82%), *Rhizophagus* (~14.74%), and *Claroideoglomus* (~0.43%). Minor contributions were observed from *Sclerocystis* (~0.08%) and *Paraglomus* (~0.008%), with a small fraction (~0.67%) comprising unidentified genera. In terms of species richness, the identified AMF complex includes *Glomus* spp., *Rhizophagus intraradices*, *Rhizophagus clarus*, *Sclerocystis sinuosa*, and *Paraglomus majewskii*, alongside several unidentified species within the genera *Glomus* and *Paraglomus*.

### 3.2. Mycorrhizal Frequency and Intensity in Pea Seedlings Following Various Treatments

Table 1 summarizes the mycorrhizal frequency (F%) and intensity (M%) in *Pisum sativum* seedlings exposed to various treatments, including the control, *Bacillus pumilus* (PGPB), arbuscular mycorrhizal fungi (AMFs), and the combined inoculation of PGPB + AMFs. Mycorrhizal colonization was undetectable in both the control and PGPB treatments, with an F% of 0.0 ± 0.0%. Conversely, AMF and PGPB + AMF treatments resulted in complete colonization, each exhibiting an F% of 100 ± 0.0%. These results clearly demonstrate that AMF inoculation, either alone or in combination with PGPB, effectively establishes mycorrhizal symbiosis in *Pisum sativum* seedlings. Regarding the mycorrhizal intensity (M%), no significant colonization was observed after the control and PGPB treatments (M% = 0.0 ± 0.0%). The AMF treatment resulted in a mycorrhizal intensity of 36.25 ± 2.50%, whereas the combined PGPB + AMF treatment slightly increased the intensity to 41.25 ± 4.79% (Table 1). However, this difference was not statistically significant.

### 3.3. Morphological and Biochemical Characterization of the Bacillus Pumilus Strain

The *Bacillus pumilus* (*B. pumilus*) strain isolated to promote *Pisum sativum* growth exhibited various distinctive morphological and biochemical characteristics (Figure 1). Macroscopically, the bacterium was observed to form round, beige-colored colonies that could reach up to 6.5 mm in diameter (Table 2). Microscopically, the strain was identified as Gram-positive, spore-forming, and mobile, with the ability to diffuse on a mannitol-mobility agar medium (Table 2), which confirmed its motility. The bacterial cells were rod-shaped (bacilli), with parallel edges and round ends, consistent with typical characteristics of the *Bacillus* genus.

Biochemically, the strain exhibited facultative aerobic metabolism and was capable of hydrolyzing starch, reducing nitrates, and degrading arginine. It also demonstrated the ability to utilize several sugars, including fructose, maltose, and mannose, as carbon sources and tested positive for gelatinase activity and catalase activity. The latter was confirmed by the production of bubbles upon the addition of hydrogen peroxide, indicating the presence of catalase, an enzyme responsible for the breakdown of hydrogen peroxide (Table 3). Furthermore, the strain was capable of nitrogen fixation, as evidenced by its robust growth on nitrogen-free media within 48 h at 30 °C. This capacity highlights the strain’s potential to contribute to nitrogen availability in the soil, a key factor in promoting plant growth.

Regarding phytohormone production, the *B. pumilus* strain was found to synthesize indole-3-acetic acid (IAA) at a concentration of 1.055 µg·mL^−1^ when grown in Luria-Bertani medium supplemented with L-tryptophan (1 g/L), indicating its potential to enhance plant growth through hormone production. The production of hydrogen cyanide (HCN) was also demonstrated (Table 4), with the strain exhibiting a color change on Whatman paper from yellow to dark brown, confirming the production of this metabolite on a glycine-supplemented agar medium (4.4 g/L) after an incubation at 28 °C for four days.

In summary, the *B. pumilus* strain demonstrated several key traits contributing to its potential as a plant growth-promoting bacterium, including nitrogen fixation, IAA production, HCN production, and enzymatic activities like catalase and gelatinase activities [36]. These findings underscore the strain’s suitability for use as a bioinoculant to enhance the growth and yield of *Pisum sativum*. The detailed morphological and biochemical profiles are summarized in Table 1, Table 2, Table 3 and Table 4. However, the strain did not exhibit phosphate solubilization, as determined by the method outlined by Gupta et al. [37], indicating that phosphate solubilization is not a characteristic feature of this isolate.

### 3.4. Demonstration of the Stimulation of the Growth Metrics of Peas

One-way ANOVA revealed that biostimulant treatments significantly influenced all four measured growth parameters: plant height, collar diameter, leaf production, and flower/pod production (Table 5).

### 3.5. Plant Height

The control group, consisting of non-inoculated plants, exhibited the shortest height (56.75 ± 4.57 cm). Plants inoculated with the *Bacillus pumilus* (*B. pumilus*) strain (PGPB) showed a significant increase in height, reaching 79.00 ± 1.15 cm, corresponding to a ~39.2% improvement over the control. Similarly, plants inoculated with a mixture of indigenous arbuscular mycorrhizal fungi (AMFs) achieved a height of 80.25 ± 3.40 cm, representing a ~41.4% increase over the control. The combined treatment, where plants were inoculated with both the *B. pumilus* strain and the indigenous AMF mixture (PGPB + AMFs), produced the tallest plants (86.50 ± 4.04 cm), resulting in a significant ~52.5% improvement compared to the control (Figure 2A).

### 3.6. Collar Diameter

The collar diameter of plants was significantly increased following all biostimulant treatments compared to the control group (6.25 ± 0.50 mm), which consisted of non-inoculated plants. Inoculation with the *Bacillus pumilus* strain (PGPB) increased the collar diameter to 8.75 ± 0.50 mm, representing a ~40.0% improvement over the control. Plants inoculated with the indigenous arbuscular mycorrhizal fungi (AMFs) achieved a collar diameter of 9.25 ± 0.50 mm, reflecting a ~48.0% increase. The most pronounced effect was observed for the combined treatment (PGPB + AMFs), where plants exhibited the largest collar diameter (11.00 ± 0.82 mm), corresponding to a ~76.0% improvement over the control (Figure 2B).

### 3.7. Leaf Production

All biostimulant treatments significantly enhanced leaf production. Control plants produced an average of 85.50 ± 7.33 leaves per plant. Inoculation with PGPB alone increased the leaf number to 94.25 ± 4.79, representing a ~10.2% improvement over the control. Plants treated with AMFs alone produced 95.25 ± 2.50 leaves per plant, showing an ~11.4% improvement. The combined treatment (PGPB + AMFs) resulted in the highest leaf production (99.68 ± 2.01 leaves per plant), representing a ~16.6% increase compared to the control and outperforming the individual treatments (Figure 2C).

### 3.8. Flower and Pod Production

Biostimulant treatments significantly enhanced both flower and pod production, with the most notable effects observed for the combined treatment. Control plants produced the fewest flowers and pods (23.75 ± 4.65). Inoculation with PGPB alone increased the number of flowers and pods to 30.25 ± 3.86, reflecting a ~27.4% improvement over the control. Treatment with AMFs alone resulted in 37.00 ± 2.58 flowers and pods, corresponding to a ~55.8% improvement. The combined treatment (PGPB + AMFs) produced the highest number of flowers and pods (41.25 ± 3.40), representing a statistically significant ~73.6% increase over the control and surpassing the effects of the individual treatments (Figure 2D).

### 3.9. Biomass Production of Peas

In all instances, the combined PGPB + AMF treatment consistently resulted in the highest biomass values across all measured parameters, demonstrating a synergistic effect of these biostimulants on plant growth. The control group (non-inoculated) exhibited the lowest values for shoot and root fresh and dry weights, as well as total fresh and dry weights. Specifically, the shoot fresh weight (SFW) of control plants was 27.25 ± 1.26 g, which was substantially lower than that of the PGPB + AMF-treated plants, which exhibited a significant increase to 46.25 ± 0.29 g, reflecting a ~69.7% improvement. Similarly, the shoot dry weight (SDW) of control plants was 13.25 ± 0.96 g, while the PGPB + AMF treatment resulted in a marked increase to 19.00 ± 0.00 g, a ~43.6% enhancement. The root fresh weight (RFW) of control plants was 13.63 ± 0.63 g, and this was significantly improved by all treatments, with the combined PGPB + AMF inoculation achieving an RFW of 22.75 ± 1.19 g, corresponding to a ~66.8% increase. The root dry weight (RDW) of the control plants was 4.50 ± 0.38 g, and this trait was also significantly enhanced by the combined treatment, which resulted in an RDW of 7.78 ± 0.14 g, a ~72.7% improvement. The total fresh weight (TFW) of control plants was 40.88 ± 1.89 g, which was significantly lower than that after the PGPB + AMF treatment, which increased the TFW to 69.00 ± 1.38 g, reflecting a ~68.7% increase. The total dry weight (TDW) of the control group was 17.75 ± 0.96 g, and the PGPB + AMF treatment significantly increased the TDW to 26.60 ± 0.42 g, representing a ~50.0% enhancement. The statistical analysis revealed significant improvements across all treatments, with distinct groupings indicated by different letters (Figure 3). One-way ANOVA revealed that biostimulant treatments significantly influenced all measured biomass traits (*p* < 0.05) (Table 5).

### 3.10. Chlorophyll Contents of Peas

The chlorophyll a (*Chl-a*) content, chlorophyll b (*Chl-b*) content, total chlorophyll content, and the chlorophyll index (SPAD) were significantly enhanced by the biostimulant treatments applied to pea (*Pisum sativum)* plantlets (Figure 4). The combined inoculation with *Bacillus pumilus* and mycorrhizal fungi (PGPB + AMFs) consistently resulted in the highest levels of all photosynthetic pigments and SPAD index compared to the control and individual inoculation treatments. The *Chl-a* content exhibited a marked increase, rising from 0.46 mg g^−1^ FW in the control group to 0.94 mg g^−1^ FW with PGPB inoculation, effectively doubling the baseline value. AMF treatment further elevated the *Chl-a* content to 1.08 mg g^−1^ FW, representing a ~135% increase relative to the control. The most pronounced enhancement was observed for the PGPB + AMF treatment, which boosted the *Chl-a* level by ~180%, culminating in a final value of 1.29 mg g^−1^ FW. A comparable trend was evident for the *Chl-b* content, which increased from 0.37 mg g^−1^ FW in the control plants to 0.46 mg g^−1^ FW after the PGPB treatment, reflecting an improvement of over ~24%. AMF treatment further increased the *Chl-b* content to 0.48 mg g^−1^ FW, while the combined treatment led to a substantial ~75% increase, reaching 0.65 mg g^−1^ FW. The total chlorophyll (*Chl a + b*) content also showed significant improvements, rising from 0.83 mg g^−1^ FW in the control plants to 1.41 mg g^−1^ FW after the PGPB treatment, corresponding to a ~70% increase. AMF treatment resulted in an 88% enhancement, reaching 1.56 mg g^−1^ FW, while the combined PGPB + AMF treatment achieved a striking ~134% increase, culminating at 1.94 mg g^−1^ FW. The SPAD index, an indicator of the chlorophyll concentration and leaf greenness, followed a similar pattern. It increased from 31.75 in the control plants to 44.50 in PGPB-treated plants, representing a ~40% improvement. AMF treatment further elevated the SPAD index to 48.25, a ~52% increase, while the combined treatment achieved the highest value of 49.88, marking a ~57% enhancement. The statistical analysis results indicated significant differences among the treatments, with different letters signifying separate groups (Figure 4).

### 3.11. Photosynthetic Yield

One-way ANOVA revealed that biostimulant treatments significantly influenced photosystem II efficiency (F*v*/F*m*) in all treatment groups (*p* < 0.05) (Table 5). The control group (non-inoculated) exhibited a baseline F*v*/F*m* value of 0.677 ± 0.022, which was significantly lower than those of the inoculated treatment groups. Plants treated with *Bacillus pumilus* (PGPB) and arbuscular mycorrhizal fungi (AMFs) exhibited F*v*/F*m* values of 0.784 ± 0.009 and 0.785 ± 0.014, respectively, representing a ~16.0% improvement compared to the control for both treatments. The combined PGPB + AMF treatment demonstrated the highest F*v*/F*m* value of 0.802 ± 0.005, reflecting an ~18.5% increase over the control. The statistical analysis confirmed significant differences between the treatments, with distinct groupings indicated by different letters (Figure 5).

### 3.12. Multivariate Analysis of Trait Interactions in Pisum Sativum Following the Microbial Treatments

The multivariate statistical analysis of *Pisum sativum* revealed significant insights into the synergistic effects of *Bacillus pumilus* (*B. pumilus*) and a mycorrhizal fungal mixture on plant performance. The PCA revealed that the first two principal components (PCs) accounted for ~91.7% of the total variance, with PC1 and PC2 contributing ~85.75% and ~5.95%, respectively, thus capturing the majority of the variability within the dataset (Figure 6A). The PCA biplot demonstrated distinct clustering of the treatment groups, with control plants exhibiting limited variability and clustering closely in the negative region of PC1, indicative of lower performance across the measured traits. In contrast, plants treated with *B. pumilus* alone (PGPB) showed moderate improvements, separating slightly along PC1. Notably, the combined treatment of *B. pumilus* and a mycorrhizal fungal mixture (PGPB + AMFs) achieved significant separation in the positive PC1 region, reflecting substantial enhancements in biomass-related parameters (total dry weight, root dry weight, and shoot dry weight), photosynthetic efficiency (F*v*/F*m*), chlorophyll content (Chl-a, Chl-b, and total chlorophyll), and the SPAD index. The strong alignment of these traits with the positive loadings of PC1 further highlights the synergistic effects of the microbial consortium on promoting plant growth and productivity. The Pearson correlation heatmap corroborated these findings, with positive correlations observed between morphological traits (leaf number and flower number), biomass parameters, the chlorophyll content, and photosynthetic activity, reinforcing the interdependence of these variables in driving plant performance (Figure 6B). Negative or weak correlations were noted for parameters such as the moisture content, suggesting possible trade-offs or environmental influences on resource allocation.

## 4. Discussion

This study isolates and characterizes a *Bacillus pumilus* (*B. pumilus*) strain with the potential to promote the growth of *Pisum sativum* (*P. sativum*) and examines its combined effects with a mycorrhizal fungal complex on plant growth, yield, and photosynthetic performance in a greenhouse setting. The combination of *B. pumilus* and arbuscular mycorrhizal fungi (AMFs) resulted in a significant enhancement of plant growth, supporting the hypothesis that microbial synergism plays a crucial role in improving crop productivity. Our results provide new insights into how *B. pumilus*—a bacterium known for its versatility in nitrogen fixation, indole-3-acetic acid (IAA) production, and pathogen suppression—can enhance plant–microbe interactions, specifically in conjunction with the AMF complex. These findings align with previous studies that demonstrate the positive effects of PGPRs and AMFs on plant health [38,39,40]. However, our study expands on this by elucidating the specific mechanisms underlying these effects. The phenotypic and biochemical characterization of *B. pumilus* in our study revealed that the strain produces beige, spherical, elevated mucoid colonies with a diameter of up to 6.5 mm, and it is motile, catalase-positive, and nitrogen-fixing, consistent with previous reports [38]. This bacterium is found in diverse environments, including marine waters, deep-sea sediments, and soil [41], and is recognized for its robust biological functions, including the synthesis of auxins like IAA and nitrogen fixation via nitrogenase activity [42,43]. These traits align with our results, showing enhanced root development and nutrient uptake in *P. sativum* upon inoculation with *B. pumilus*. Our findings suggest that IAA production plays a significant role in improving the root architecture, facilitating greater nutrient absorption, and contributing to overall plant growth, consistent with earlier reports [11]. The bacterium’s nitrogen-fixing ability likely further enhances soil fertility, promoting healthier plants through improved nitrogen availability [44]. The versatility of *B. pumilus* was further evidenced by its cellulase activity, which contributes to the breakdown of cellulose into simpler sugars, enhancing nutrient availability, particularly in nutrient-poor soils [45]. Additionally, the bacterium’s ability to degrade starch, fructose, mannose, and maltose further supports its capacity to improve soil fertility and plant growth, especially under challenging conditions [42,43]. Our data suggest that these biochemical activities, combined with AMFs, create a beneficial environment that enhances plant health by improving nutrient access, particularly in soils with limited nutrients. Moreover, the biocontrol potential of *B. pumilus* through hydrogen cyanide (HCN) production adds another layer of benefit. HCN is known to inhibit the growth of soil-borne pathogens, contributing to a healthier rhizosphere [46]. This biocontrol activity, coupled with the motility of *B. pumilus*, enables it to effectively colonize plant roots, thereby providing better protection against pathogens and environmental stressors [47]. The bacterium’s ability to enhance antioxidant enzyme activity, such as catalase production, likely mitigated oxidative stress, a significant factor involved in plant growth under stress conditions [38,48]. This capacity for reducing oxidative stress is a crucial attribute, particularly in agricultural environments where abiotic stresses such as drought and nutrient limitations frequently occur. Overall, the combination of beneficial traits such as motility, IAA production, cellulase activity, nitrogen fixation, HCN production, and environmental stress resilience makes *B. pumilus* a highly effective PGPR. These traits support plant growth, enhance soil fertility, suppress pathogens, and boost stress resistance, positioning *B. pumilus* as a promising candidate for improving agricultural productivity and sustainability.

The dual inoculation of *P. sativum* plantlets with *B. pumilus* and a mixture of mycorrhizal fungi led to significant improvements across a range of plant growth parameters, including morphological traits, biomass production, chlorophyll content, and photosynthetic efficiency. These findings are consistent with previous research that has demonstrated the synergistic effects of combining PGPRs with AMFs, which enhance plant growth by optimizing nutrient uptake and improving physiological processes [20,49,50,51,52,53,54,55]. The impact of *B. pumilus* on plant growth has been well-documented in the literature, with numerous studies highlighting its positive effects on a variety of plant species. As a plant growth-promoting bacterium, *B. pumilus* enhances plant development through the production of phytohormones and growth-promoting chemicals [56]. For example, inoculating *Oryza sativa* (rice) plants with *B. pumilus* resulted in significant improvements in root surface area, length, and node count, along with increased chlorophyll and nitrogen contents compared to control plants [38]. Similarly, inoculating *Camellia sinensis* (Chinese tea) plants with *B. pumilus* led to favorable outcomes, including enhanced plant height and leaf production [57]. In wheat (*Triticum aestivum*, var. Orkhon), *B. pumilus* inoculation resulted in an increased dry weight and nitrogen content [58]. In legumes, particularly lentils (*Lens culinaris* Medik.), inoculation with *B. pumilus* significantly boosted plant length and fresh weight [59]. Additionally, inoculating *Phaseolus vulgaris* (beans) with *B. pumilus* WP8 led to greater shoot length, aboveground dry weight, root length, and root dry weight [60]. These studies support our observation that *B. pumilus* enhances plant growth through similar mechanisms, likely through improved nutrient acquisition, hormonal modulation, and enhanced root development. Advantageously, studies involving microbial consortia, such as the combination of *Bacillus* spp. and AMFs, have shown their potential to improve plant growth and stress tolerance. Costa-Santos et al. [61] reported that *Bacillus* strains enhanced the growth of *Solanum lycopersicum* (tomato), particularly in adult plants, by increasing pigment content and photosynthetic efficiency. Additionally, Awasthi et al. [62] demonstrated that the combination of AM spores, i.e., *Glomus mosseae* and *Bacillus subtilis* (a nitrogen-fixing bacterium), significantly enhanced the biomass yield, nutrient uptake, and artemisinin content in *Artemisia annua*. These findings mirror our results, suggesting that similar microbial consortia could benefit *P. sativum* by promoting both primary and secondary metabolite production. For example, co-inoculation with *Bacillus amyloliquefaciens* and AMFs enhanced soybean biomass, yield, and seed quality under drought stress. This treatment improved the levels of primary metabolites, antioxidants, osmoprotectants, and stress-related hormones, boosting stress tolerance and overall plant performance [63]. Similarly, the co-inoculation of *Spinacia oleracea* plants with mycorrhizal fungi and bacteria enhanced root colonization and significantly increased the concentrations of total phenolic compounds, flavonoids, and phenolic acids. This treatment also improved antioxidant activity, particularly through increased quercetin and chlorogenic acid levels, and enhanced the plant chlorophyll content [64]. In the context of abiotic stress, recent studies on salt-affected soils have highlighted microbial consortia’s potential to alleviate plant stress. Moreira et al. [65] found that a combination of PGPR strains (*Pseudomonas reactants* EDP28 and *Pantoea alli* ZS 3-6) and an AMF (*Rhizoglomus irregulare*) significantly improved maize growth and the nutritional status under saline stress. The dual inoculation enhanced K^+^ uptake and reduced Na^+^ accumulation, mitigating ion imbalances across a salinity gradient (0–5 g NaCl kg^−1^). This aligns with our findings, which suggest that microbial interactions in *P. sativum* can improve growth and stress tolerance, potentially through similar mechanisms of enhanced nutrient uptake and stress mitigation. Our results showed significant improvements in plant height (~52.5%), collar diameter (~76%), and leaf production (~16.6%) in the dual inoculation treatment group, consistent with studies that report increases in leguminous plants’ size and growth when PGPRs are paired with AMFs [7,66,67,68,69]. These findings highlight the roles of these microbes in improving nutrient availability and root development. Additionally, Nanjundappa et al. [48] emphasized the dual roles of *Bacillus* spp. and AMFs in enhancing plant vigor, chlorophyll synthesis, and photosynthesis—key factors for improving growth under suboptimal conditions. Notably, flower and pod production increased by ~73.6% in the combined treatment group compared to the control group, suggesting enhanced reproductive success, which is crucial for higher yields. Zeng et al. [9] also observed that a dual microbial inoculation improved not only vegetative growth but also reproductive performance, likely due to enhanced nutrient uptake and hormonal modulation. For example, a study by Khan and Zaidi [66] found that the triple inoculation of *Azotobacter chroococcum*, *Bacillus* spp., and *Glomus fasciculatum* significantly increased wheat growth and grain yield. The combination of these microbes resulted in a two-fold increase in grain yield and a substantial increase in the nitrogen and phosphorus contents, underscoring the potential of microbial consortia to improve nutrient acquisition and overall crop productivity. The most pronounced improvements in biomass production occurred with the dual inoculation, leading to a ~69.7% increase in shoot fresh weight and a ~72.7% increase in root dry weight. These enhancements can be attributed to the increased nutrient uptake facilitated by both microbes. *B. pumilus*, known for its nitrogen-fixing and nutrient-solubilizing properties, contributes to the availability of essential nutrients like nitrogen and phosphorus, thereby improving plant health and growth [8,70,71]. On the other hand, mycorrhizal fungi are crucial for enhancing the uptake of phosphorus, nitrogen, and other micronutrients, which are often limiting in soils [19,24]. Together, these microbes promote efficient resource use, leading to improved growth and a higher biomass, which is consistent with several studies that showed that AMFs significantly enhance nutrient acquisition in crops [22,23,72,73,74].

In terms of photosynthetic capacity, we observed a substantial increase in the chlorophyll content, with chlorophyll a, b, and total chlorophyll levels increasing by ~180%, ~75%, and ~134%, respectively, after the combined inoculation. This increase was accompanied by a ~18.5% higher photosynthetic yield (F*v*/F*m*) in inoculated plants, a key indicator of an enhanced energy conversion capacity. The multivariate analysis further supported these findings, revealing that the dual inoculation group was distinctly separated from the control group and other treatment groups along the first principal component, explaining ~85.75% of the variance in the dataset. This suggests that the dual microbial treatment resulted in comprehensive improvements across all measured traits, confirming the synergistic effects of *B. pumilus* and AMFs on the growth and productivity of *P. sativum*. These findings are consistent with Nader et al. [75], who reported that dual inoculation with *Bacillus* spp. and AMFs led to a higher chlorophyll content and photosynthetic efficiency in crops like soybean, contributing to improved growth and biomass production. Shaffique et al. [16] demonstrated that the *B. pumilus* SH-9 strain significantly enhances soybean plant growth and biomass under drought stress. This effect is attributed to its ability to promote phosphate solubilization and siderophore production and strengthen antioxidant defense systems while simultaneously reducing abscisic acid levels and upregulating key transcription factors. These actions improve the plant’s physiological traits, such as the chlorophyll content and water retention. Similarly, *B. pumilus*, isolated from *Artemisia vulgaris*, exhibits promising potential for stress tolerance and plant growth promotion by expressing genes linked to phytohormone biosynthesis and stress-resilient traits [76]. Furthermore, the SH-9 strain also improves rice seed germination and seedling growth under drought conditions by producing phytohormones and antioxidants, and solubilizing phosphate. This strain’s drought tolerance and plant growth-promoting capabilities position it as a promising candidate for sustainable agricultural practices, particularly through seed biopriming [15]. Yadav et al. [77] showed that the *Bacillus subtilis* CP4 isolate, combined with AMFs, significantly enhanced biofortification and yield under field conditions, with upregulated metabolites linked to improved plant growth and nutrient uptake.

Overall, the current study supports the potential of combining *B. pumilus* and mycorrhizal fungi as an effective strategy for enhancing the growth, biomass, and yield of *P. sativum*. The dual inoculation not only improved key growth parameters but also enhanced the photosynthetic capacity and chlorophyll content, which could lead to an increased overall crop yield. These results are in line with a growing body of research advocating for microbial consortia as a promising tool in sustainable agriculture. Further studies could explore the molecular mechanisms behind this synergy, particularly the roles of microbial metabolites and their interactions with plant physiological responses, offering valuable insights for optimizing microbial applications in agriculture.

## 5. Conclusions

This study highlights the significant roles of plant growth-promoting rhizobacteria (PGPR) and mycorrhizal fungi in enhancing plant growth and agricultural productivity. We focused on isolating *Bacillus pumilus* from the rhizosphere of *Acacia cyanophylla*, a strain exhibiting a range of beneficial biochemical activities, including nitrogen fixation, indole-3-acetic acid production, ammonia and hydrogen cyanide synthesis, as well as the secretion of key enzymes, such as catalase, cellulase, and gelatinase. The growth-promoting effects of this strain, in combination with a mycorrhizal fungal complex, were evaluated on *Pisum sativum* (pea). Our findings demonstrated significant improvements in both the biomass and yield of pea plants under greenhouse conditions when treated with these dual inoculants. The combined actions of *Bacillus pumilus* and the mycorrhizal fungi resulted in a synergistic enhancement of plant growth, outperforming the untreated control plants. These results underscore the potential of microbial inoculants as a sustainable alternative to conventional chemical inputs in agriculture, offering a pathway to more eco-friendly, efficient crop production systems.

## Figures and Tables

**Figure 1 plants-14-01991-f001:**
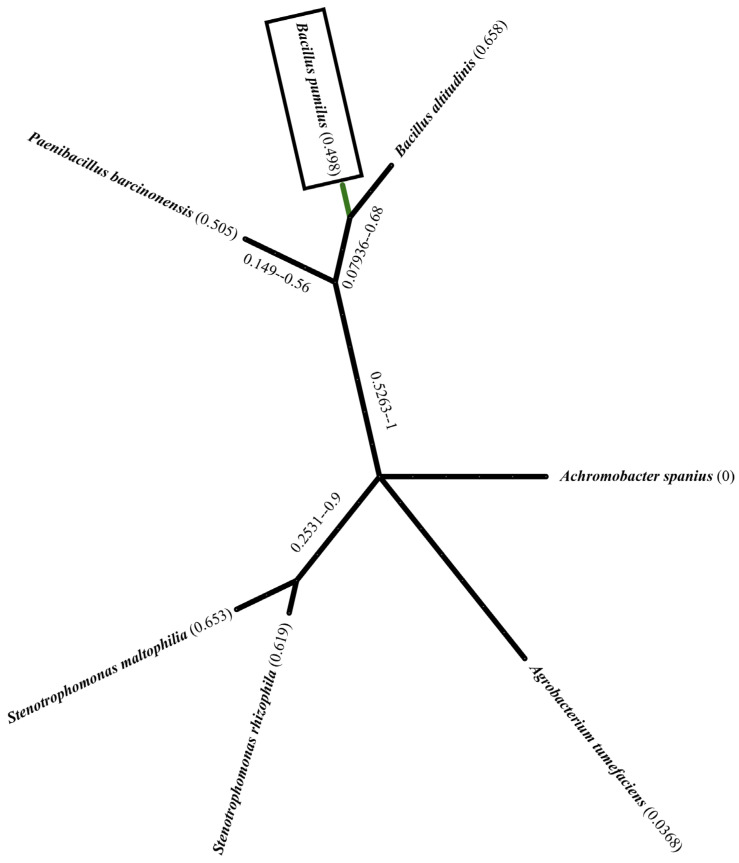
Phylogenetic tree of bacterial strains isolated from the rhizosphere soil of *Acacia cyanophylla*. It represents the phylogenetic relationships between the isolated strains and closely related species. The main focus of this study, *Bacillus pumilus*, is highlighted with a rectangular box. The tree includes other bacterial taxa such as *Bacillus altitudinis*, *Paenibacillus barcinonensis*, *Stenotrophomonas* spp., *Achromobacter spanius*, and *Agrobacterium tumefaciens*. The branch thickness reflects the evolutionary distance.

**Figure 2 plants-14-01991-f002:**
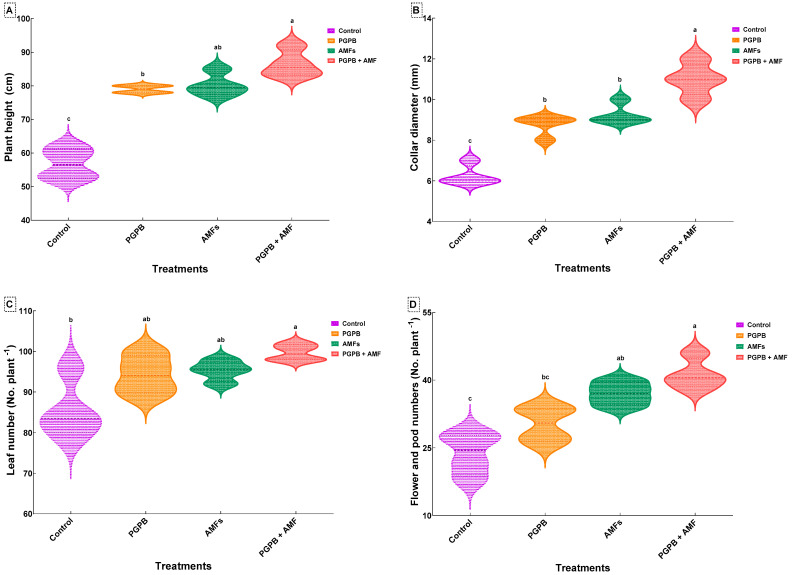
Effects of biostimulant treatments on the growth and reproductive traits of *Pisum sativum.* Violin plots showing the effects of the biostimulant treatments on plant height (**A**), collar diameter (**B**), leaf number (**C**), and flower number (**D**) in *Pisum sativum*. Plots illustrate the data distribution and density (means ± SDs, *n* = 4). Means that share the same lowercase letter within a panel are not significantly different (Tukey’s HSD test, *p* < 0.05). Treatments include the non-inoculated control (Control), inoculation with *B. pumilus* (PGPB), inoculation with mycorrhizal fungi (AMFs), and combined inoculation with *B. pumilus* and mycorrhizal fungi (PGPB + AMFs).

**Figure 3 plants-14-01991-f003:**
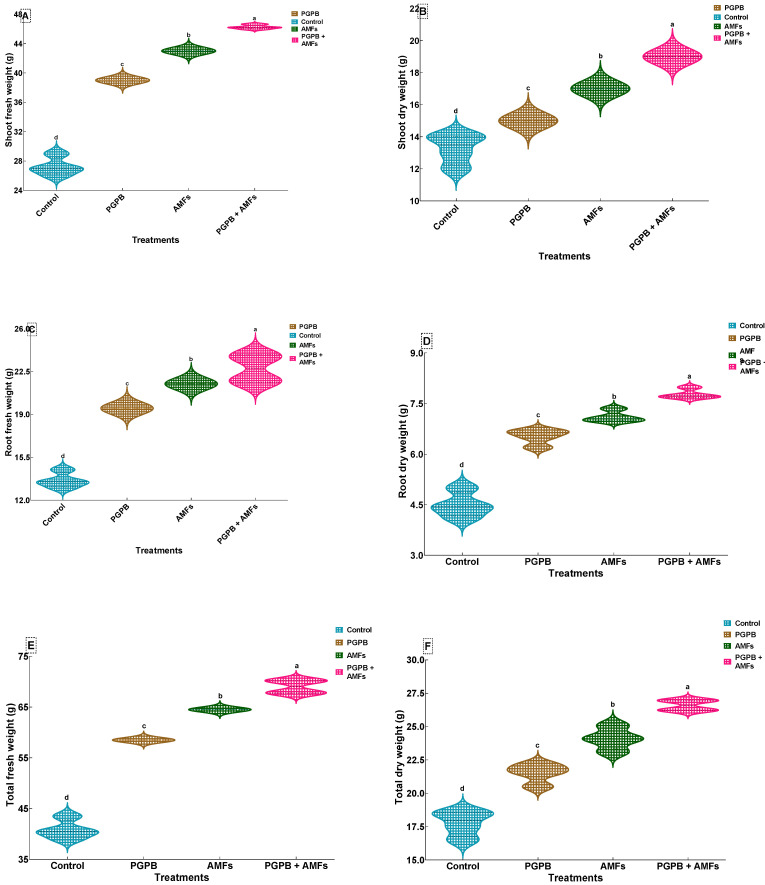
Effects of biostimulant treatments on biomass production in *Pisum sativum.* Violin plots showing the effects of biostimulant treatments on shoot fresh weight (**A**), shoot dry weight (**B**), root fresh weight (**C**), root dry weight (**D**), total fresh weight (**E**), and total dry weight (**F**) of *Pisum sativum*. Violin plots illustrate the distribution and density of data (means ± SDs, *n* = 4). Means sharing the same lowercase letter within each panel are not significantly different according to Tukey’s HSD test (*p* < 0.05). Treatments include a non-inoculated control (Control), inoculation with *B. pumilus* (PGPB), inoculation with a mixture of mycorrhizal fungi (AMFs), and combined inoculation with *B. pumilus* and mycorrhizal fungi (PGPB + AMFs).

**Figure 4 plants-14-01991-f004:**
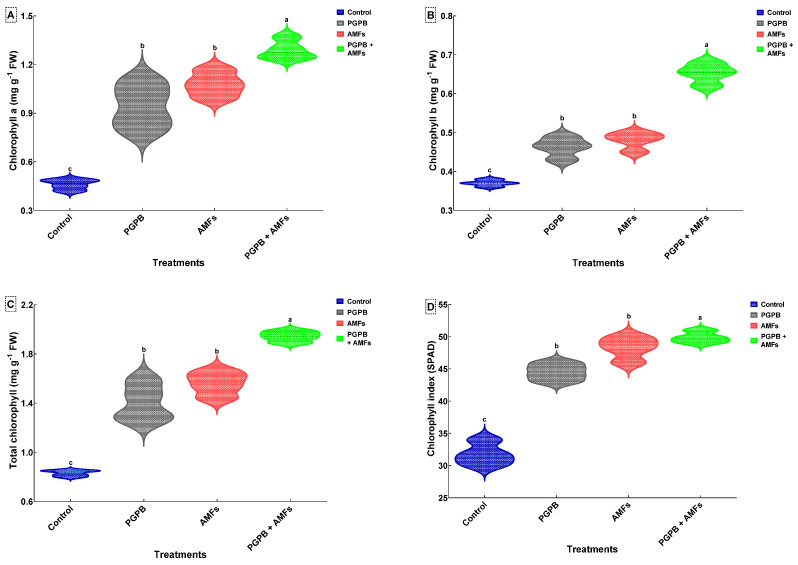
Effects of biostimulant treatments on the chlorophyll content and SPAD index in *Pisum sativum.* Violin plots depicting the effects of biostimulant treatments on the chlorophyll a content (**A**), chlorophyll b content (**B**), total chlorophyll (*Chl a + b*) content (**C**), and chlorophyll index (SPAD) (**D**) in *P. sativum*. Each plot illustrates the distribution and density of data (means ± SDs, *n* = 4). Treatments include the non-inoculated control (Control), inoculation with *B. pumilus* (PGPB), inoculation with mycorrhizal fungi (AMFs), and combined inoculation with *B. pumilus* and mycorrhizal fungi (PGPB + AMFs). Means sharing the same lowercase letter within a panel are not significantly different (Tukey’s HSD test, *p* < 0.05).

**Figure 5 plants-14-01991-f005:**
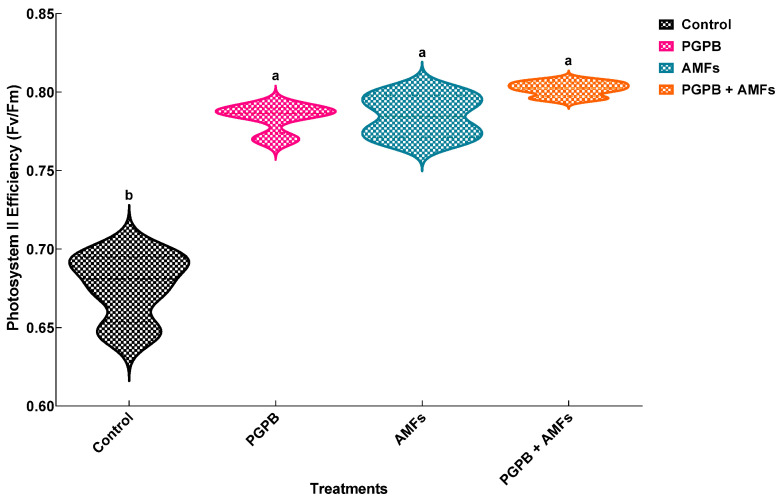
Effects of biostimulant treatments on photosynthetic yield in *Pisum sativum.* Violin plots depicting the effects of biostimulant treatments on photosystem II efficiency (F*v*/F*m*) in *Pisum sativum*. The plot illustrates the distribution and density of data (means ± SDs, *n* = 4). Treatments included a non-inoculated control (Control), inoculation with *B. pumilus* (PGPB), inoculation with mycorrhizal fungi (AMFs), and combined inoculation with *B. pumilus* and mycorrhizal fungi (PGPB + AMFs). Means sharing the same lowercase letter within a panel are not significantly different (Tukey’s HSD test, *p* < 0.05).

**Figure 6 plants-14-01991-f006:**
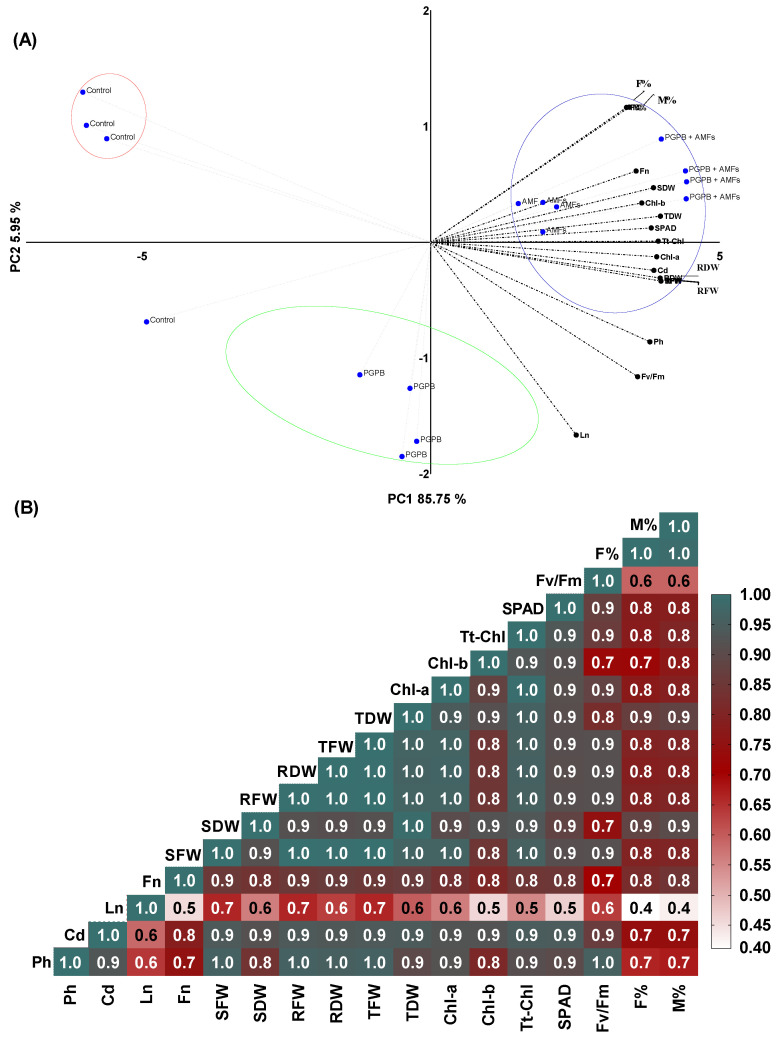
Multivariate analysis of the morphological, biomass, chlorophyll, photosynthetic, and yield traits of *Pisum sativum*: principal component analysis (**A**) and Pearson’s correlation heatmap (**B**). Ph = plant height (cm); Cd = collar diameter (mm); Ln = leaf number; Fn = flower number; SDW = shoot dry weight; SFW = shoot fresh weight; RDW = root dry weight; RFW = root fresh weight; TDW = total dry weight; TFW = total fresh weight; Chl-b = chlorophyll b; Chl-a = chlorophyll a; SPAD = chlorophyll index; Tt-Chl = total chlorophyll (a + b); F*v*/F*m* = photosynthetic efficiency; F% = mycorrhizal frequency; M% = mycorrhizal intensity.

**Table 1 plants-14-01991-t001:** Mycorrhizal frequency and mycorrhizal intensity in *Pisum sativum* seedlings under different treatment conditions.

	Control	PGPB	AMFs	PGPB + AMFs
F (%)	0.0 ± 0.0 ^b^	0.0 ± 0.0 ^b^	100 ± 0.0 ^a^	100 ± 0.0 ^a^
M (%)	0.0 ± 0.0 ^b^	0.0 ± 0.0 ^b^	36.25 ± 2.50 ^a^	41.25 ± 4.79 ^a^

The values are the means ± SDs (*n* = 4). For each parameter, means followed by the same lowercase letter within a row are not significantly different according to Tukey’s HSD test (*p* < 0.05). Treatments include the non-inoculated control (Control), inoculation with *B. pumilus* (PGPB), inoculation with arbuscular mycorrhizal fungi (*AMFs*), and combined inoculation with *B. pumilus* and *AMFs* (PGPB + AMFs).

**Table 2 plants-14-01991-t002:** Macroscopic and microscopic characteristics of the *Bacillus pumilus* strain isolated to promote the growth of *Pisum sativum*.

Macroscopic Characteristics						
Odor	Form	Color	Margin	Elevation	Surface	Size (mm)
Odorless	Round	Cream	Whole	Raised	Mucoid	6.5
Microscopic Characteristics						
Gram reaction		Form	Motility		Sporulation	
Positive (+)		*Bacillus*	Mobile		Positive (+)	

**Table 3 plants-14-01991-t003:** Biochemical identification of the *Bacillus pumilus* strain isolated to promote the growth of *Pisum sativum*.

Biochemical Characteristic	*Bacillus pumilus*
Breathing type	Optional Aerobics
Csein	-
NaCl	-
Hydrolysis of starch	+
Growth at 50 °C	-
Growth at 60 °C	-
Nitrate reduction	+
Mannose utilization	+
Arabinose utilization	-
Fructose utilization	+
Maltose utilization	+
Citrate utilization	-
Arginine degradation	+
Gelatinase activity	+
Catalase activity	+

**Table 4 plants-14-01991-t004:** Biochemical activity of the *Bacillus pumilus* strain isolated to promote the growth of *Pisum sativum*.

Strains	AIA Production	PO4 Solubilization	NH3 Production	Hydrogen Cyanide Production	Pectinase Activity	Cellulase Activity	N Fixation Activity
*B. pumilus*	+	-	+	+	-	+	+

**Table 5 plants-14-01991-t005:** ANOVA results for the morphological, biomass, chlorophyll, photosynthetic, and yield traits of *Pisum sativum*.

Parameters	DF	F	ANOVA
Plant height (cm)	3	53.93	<0.0001
Collar diameter (mm)	3	43.47	<0.0001
Leaves number	3	6.487	0.0074
Flower and pod numbers	3	17.26	0.0001
Shoot fresh weight	3	661.2	<0.0001
Soot dry weight	3	107.9	<0.0001
Root fresh weight	3	144.1	<0.0001
Root dry weight	3	133.2	<0.0001
Total fresh weight	3	445.5	<0.0001
Total dry weight	3	99.46	<0.0001
Chlorophyll a	3	68.89	<0.0001
Chlorophyll b	3	124.6	<0.0001
Total chlorophyll (*Chl a + b*)	3	100.9	<0.0001
Chlorophyll index (SPAD)	3	131.1	<0.0001
Photosynthetic efficiency (F*v*/F*m*)	3	67.34	<0.0001
F (%)	3	273.8	<0.0001
M (%)	3	276.9	<0.0001

## Data Availability

All data generated or analyzed during this study are included in this article.

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
