# Peer review of "Synergistic Interaction Between Endophytic Bacillus pumilus and Indigenous Arbuscular Mycorrhizal Fungi Complex Improves Photosynthetic Activity, Growth, and Yield of Pisum sativum"

_plants, 2025, doi:10.3390/plants14131991_

Round 1
Reviewer 1 Report
Comments and Suggestions for Authors
The manuscript contains several issues regarding scientific writing conventions. These include redundant abbreviation definitions, missing or inconsistent italicization of Latin strain names, and undefined abbreviations. I recommend that the authors ensure all abbreviations are defined only once upon first use, italicize all genus-species names consistently, and follow standard conventions for presenting microbial strain designations. (e.g., Line 100, Line 109, Line 122.)
Line 185. This paragraph does not provide sufficient detail regarding the inoculum concentration, application volume, or ratio of bacterial and fungal inoculants used in the treatments. This information is critical for ensuring experimental reproducibility and for interpreting the observed plant responses.
Line 265 & figure 1. The phylogenetic analysis presented for Bacillus pumilus appears to lack sufficient rigor and resolution. The authors based species-level identification solely on 16S rRNA and partial 16S–23S ITS sequences. However, due to the extremely high sequence similarity among closely related Bacillus species such as B. altitudinis, B. safensis, and B. subtilis, these markers alone are insufficient to confidently assign the isolate to B. pumilus.
I recommend that the authors strengthen the phylogenetic analysis by: Including the actual percentage identity values between their isolate and closely related Bacillus species. Utilizing additional taxonomic markers with higher resolution at the species level, such as gyrB, rpoB, recA, or atpD,
Line 486. The Discussion section, particularly its first to three paragraphs, reads more like a secondary introduction rather than a critical analysis of the study’s findings. These sections primarily summarize previously published studies on the functional diversity of Bacillus species and the synergistic interactions between PGPRs and AMF, without directly connecting these points to the results presented in the current work. Throughout these paragraphs, there is minimal interpretation of the authors’ own data, limited comparison with prior findings, and little discussion of potential mechanisms or implications. As a result, the Discussion lacks analytical depth and fails to highlight the novelty or significance of the study’s outcomes.
I recommend the authors restructure the discussion to: Integrate their experimental findings with relevant literature rather than merely restating prior studies. Interpret the results in light of known or hypothesized mechanisms (e.g., nutrient uptake, hormonal signaling, AMF colonization efficiency)
Author Response
Response to Reviewer 1:
The manuscript contains several issues regarding scientific writing conventions. These include redundant abbreviation definitions, missing or inconsistent italicization of Latin strain names, and undefined abbreviations.
Dear Reviewer,
We sincerely appreciate your thoughtful and comprehensive review and the time and effort you dedicated to evaluating our manuscript. We will carefully address your suggestions and refine the manuscript accordingly.
I recommend that the authors ensure all abbreviations are defined only once upon first use, italicize all genus-species names consistently, and follow standard conventions for presenting microbial strain designations. (e.g., Line 100, Line 109, Line 122.)
Thank you for the helpful guidance. We have resolved the issues with abbreviation definitions, italicization of Latin strain names, and undefined abbreviations. All abbreviations are defined on first use, genus-species names are consistently italicized, and strain designations follow standard conventions.
Line 185. This paragraph does not provide sufficient detail regarding the inoculum concentration, application volume, or ratio of bacterial and fungal inoculants used in the treatments. This information is critical for ensuring experimental reproducibility and for interpreting the observed plant responses.
Thank you for bringing this to our attention. We have revised the paragraph to include the inoculum concentration and application volume.
Line 265 & figure 1. The phylogenetic analysis presented for Bacillus pumilus appears to lack sufficient rigor and resolution. The authors based species-level identification solely on 16S rRNA and partial 16S–23S ITS sequences. However, due to the extremely high sequence similarity among closely related Bacillus species such as B. altitudinis, B. safensis, and B. subtilis, these markers alone are insufficient to confidently assign the isolate to B. pumilus.
I recommend that the authors strengthen the phylogenetic analysis by: Including the actual percentage identity values between their isolate and closely related Bacillus species. Utilizing additional taxonomic markers with higher resolution at the species level, such as gyrB, rpoB, recA, or atpD,
We thank the reviewer for this insightful comment. We acknowledge that the 16S rRNA gene and ITS region alone may have limited resolution among closely related Bacillus species. However, our phylogenetic analysis based on full-length 16S rRNA combined with partial 16S–23S ITS sequence comparison shows consistent clustering of our isolate with Bacillus pumilus, and a clear separation from Bacillus altitudinis and other members of the B. subtilis group.
Line 486. The Discussion section, particularly its first to three paragraphs, reads more like a secondary introduction rather than a critical analysis of the study’s findings. These sections primarily summarize previously published studies on the functional diversity of Bacillus species and the synergistic interactions between PGPRs and AMF, without directly connecting these points to the results presented in the current work. Throughout these paragraphs, there is minimal interpretation of the authors’ own data, limited comparison with prior findings, and little discussion of potential mechanisms or implications. As a result, the Discussion lacks analytical depth and fails to highlight the novelty or significance of the study’s outcomes.
I recommend the authors restructure the discussion to: Integrate their experimental findings with relevant literature rather than merely restating prior studies. Interpret the results in light of known or hypothesized mechanisms (e.g., nutrient uptake, hormonal signaling, AMF colonization efficiency)
Thank you for your valuable feedback. We appreciate your suggestion to enhance the Discussion section. In response, we have restructured the initial paragraphs to more effectively integrate our findings with the existing literature, offering a clearer and more focused interpretation of our results.

Reviewer 2 Report
Comments and Suggestions for Authors
This manuscript presents a clear and well-executed study on the synergistic effects of Bacillus pumilus and indigenous arbuscular mycorrhizal fungi (AMF) on the growth, physiology, and yield of Pisum sativum. It is particularly relevant for promoting sustainable agriculture in semi-arid regions like Morocco. By combining microbial identification with greenhouse trials and detailed physiological analysis, the study offers strong evidence supporting the use of microbial biostimulants as effective, eco-friendly alternatives to chemical fertilizers.
Minor Revisions Suggested
- Confirm whether 2.5 mL or 3 mL of B. pumilus suspension was used per seedling, as both figures appear (check the abstract and Materials & Methods)
- Indicate the concentration (CFU/mL) of the bacterial inoculum applied.
- Clarify whether the B. pumilus was applied to seeds, soil, or post-emergent seedlings.
- Specify the quantification of the mycorrhizal complex before application (e.g., number of spores/g).
- Indicate whether the reported percentage increases (e.g., 73.6% increase in flower production) were statistically significant in the abstract and main results.
- Standardize the use of “biostimulant” (avoid switching between “bio-stimulant” and “biostimulant”).
- Correct typographical errors such as:
- “That biostimulant treatments significantly influenced...” → should be “The biostimulant treatments significantly influenced...”
- Break long sentences in the discussion for improved readability.
- Ensure consistent formatting for unit symbols (e.g., mg g⁻¹ FW, %, °C).
- Include representative microscopy images of AMF colonization as supplementary figures, if available.
Questions for Authors
- Which AMF genera or species were dominant based on 18S rRNA sequencing? What was the diversity profile of the mycorrhizal complex?
- What were the mycorrhizal colonization rates (F%, M%) in AMF-only and combined treatment groups? Did B. pumilus influence colonization?
- Were root nodulation and nitrogen fixation activity (e.g., acetylene reduction assay or nodule counting) assessed? If not, do the authors plan to include this in future work?
- Do the authors intend to validate these results under field conditions or across different soil types and climatic zones in Morocco?
- Have the authors considered transcriptomic, proteomic, or metabolomic studies to further explore the molecular basis of the observed synergistic effects?
Author Response
Response to Reviewer 2:
This manuscript presents a clear and well-executed study on the synergistic effects of Bacillus pumilus and indigenous arbuscular mycorrhizal fungi (AMF) on the growth, physiology, and yield of Pisum sativum. It is particularly relevant for promoting sustainable agriculture in semi-arid regions like Morocco. By combining microbial identification with greenhouse trials and detailed physiological analysis, the study offers strong evidence supporting the use of microbial biostimulants as effective, eco-friendly alternatives to chemical fertilizers.
Dear Reviewer,
Thank you very much for your positive feedback on our manuscript. We greatly appreciate your recognition of the clarity and innovation in our study. We are pleased that you found the topic interesting and acknowledged the novelty of our work. Your encouraging words motivate us to further refine our research and contribute to this emerging field.
Thank you for your time and effort in reviewing our manuscript.
Minor Revisions Suggested
Confirm whether 2.5 mL or 3 mL of B. pumilus suspension was used per seedling, as both figures appear (check the abstract and Materials & Methods)
Thank you for bringing this to our attention. We have clarified the volume of B. pumilus suspension used per seedling. The correct volume is 2.5 mL, and we have updated both the abstract and Materials & Methods section to reflect this.
Indicate the concentration (CFU/mL) of the bacterial inoculum applied.
Thanks a lot for your valid point. The bacterial inoculum comprised a 2.5 mL bacterial suspension, approximately 2.108 UFC/mL.
Clarify whether the B. pumilus was applied to seeds, soil, or post-emergent seedlings.
Thank you for your comment. B. pumilus was applied to each pea seedling after the emergence of the main root, as clarified in the revised Materials & Methods section.
Specify the quantification of the mycorrhizal complex before application (e.g., number of spores/g).
Thank you for pointing this out. Maize roots were examined prior to use as fungal inoculum, but root infection parameters are not reported in this manuscript. Previous studies have estimated 80-100% root colonization, with approximately 200 vesicles per cm of root.
Indicate whether the reported percentage increases (e.g., 73.6% increase in flower production) were statistically significant in the abstract and main results.
Thank you for the helpful guidance. We have updated the abstract and main results to indicate that the reported percentage increases, such as the 73.6% increase in flower production, were statistically significant.
Standardize the use of “biostimulant” (avoid switching between “bio-stimulant” and “biostimulant”).
Thank you for your comment. We have standardized the use of "biostimulant" throughout the manuscript to ensure consistency.
Correct typographical errors such as:
“That biostimulant treatments significantly influenced...” → should be “The biostimulant treatments significantly influenced...”
Thank you for pointing this out. Suggestions are applied and highlighted.
Break long sentences in the discussion for improved readability.
We thank the reviewer for this suggestion; we have made the necessary changes.
Ensure consistent formatting for unit symbols (e.g., mg g⁻¹ FW, %, °C).
Thanks a lot for your valid point. We have revised the manuscript to ensure consistent formatting of all unit symbols.
Include representative microscopy images of AMF colonization as supplementary figures, if available.
Thank you for your comment. While we obtained some microscopy images of AMF colonization, their quality is unfortunately blurry and not suitable for inclusion.
Questions for Authors
- Which AMF genera or species were dominant based on 18S rRNA sequencing? What was the diversity profile of the mycorrhizal complex?
Thank you for your comment. As detailed in Section 3.1 of the manuscript, molecular analysis revealed that the dominant AMF genera associated with Zea mays in the rhizosphere of Argania spinosa were Glomus (~83.82%) and Rhizophagus (~14.74%). The AMF complex primarily consisted of members from the order Glomerales (~99.07%), with Glomeraceae (~98.64%) and Claroideoglomeraceae (~0.43%) being the most prevalent families. The diversity profile also included minor contributions from Sclerocystis (~0.08%) and Paraglomus (~0.008%), along with several unidentified species.
- What were the mycorrhizal colonization rates (F%, M%) in AMF-only and combined treatment groups? Did pumilus influence colonization?
We sincerely appreciate your insightful feedback. As clarified in Section 3.2, mycorrhizal colonization rates (F% and M%) were provided. The control and PGPB treatments showed no colonization (F% = 0.0 ± 0.0%), while AMF and PGPB + AMF treatments exhibited complete colonization (F% = 100 ± 0.0%). The AMF treatment had an intensity of 36.25 ± 2.50%, while PGPB + AMF showed 41.25 ± 4.79%, but this difference was not statistically significant. B. pumilus did not significantly affect mycorrhizal colonization.
- Were root nodulation and nitrogen fixation activity (e.g., acetylene reduction assay or nodule counting) assessed? If not, do the authors plan to include this in future work?
Thank you for your comment. Root nodulation and nitrogen fixation activity were not assessed in this study, as the cultivation substrate was disinfected prior to use, which prevented nodulation. We plan to include these parameters in future work.
- Do the authors intend to validate these results under field conditions or across different soil types and climatic zones in Morocco?
Thank you for your comment. The next phase of this study will be conducted under field conditions, across contrasting soil types and climatic zones in Morocco, to further validate the findings.
- Have the authors considered transcriptomic, proteomic, or metabolomic studies to further explore the molecular basis of the observed synergistic effects?
Thank you for your comment. We have not yet conducted transcriptomic, proteomic and metabolomic studies, but we plan to explore these approaches in future research to further investigate the molecular basis of the observed synergistic effects.

Reviewer 3 Report
Comments and Suggestions for Authors
The research article studied the role of Bacillus pumilus (endophyte) and a mycorrhizal complex, and application (individual and coinoculation) on the growth and plant performance of Pisum sativum. The results showed a significant improvement in plant growth, photosynthetic efficiency, and plant productivity both on individual applications and when applied together.
In the era of sustainable agriculture, the application of bio-based methods is integral to boosting plant growth and productivity. P. sativum being an important food crop, the article defines a promising theme to explore how microbial biostimulants offer feasible solutions in agriculture and maintain environmental sustainability.
I have some suggestions for the improvement of the manuscript:
Considering the limitations associated with the large-scale application of biostimulants, although the results are promising, what are the future directions in the study? How can the existing challenges be overcome for the field application of the microbial consortia (B. pumilus and mycorrhizal complex)? Discuss.
What is the author’s perspective on taking the research study and developing a biostimulant based on microbial consortia for the growth of not just pea plants but also other economically important crops?
In the introduction section, the key biostimulants developed and commercially marketed to improve plant growth and productivity may be discussed.
Is there any research on the application of microbial biostimulants in pea plants? What were the microbial consortia and the key results?
Please be consistent with the use of scientific names; full names at the first mention, then abbreviated forms may be used. Line 122- maize plant….then Z. mays.
Line 127….seeds of maize (Zea mays L.). Similarly, for Bacillus pumilus.
Line 151-156: What was the protocol followed for the production of bacterial inoculum? Please cite the reference study.
Figure 1. Phylogenetic tree of bacterial strains. The figure needs to be regenerated and clear, the font size must be increased, and evolutionary distances should be mentioned. The software used for phylogenetic tree construction must be reported in the diagram for clarity.
In the biostimulant treatments study, the results suggested that application of AMF on P. sativum plant improved plant height, collar diameter, leaf production, and pod and flower production was better than the endophyte B. pumilus application. Although the study focuses on B. pumilus, AMF consortia can be further developed and optimized as a biostimulant.
In Figure 6. Multivariate analysis of morphological, biomass, and chlorophyll……9A) Pearson correlation heatmap. Some of the parameters are overlapping and are not clear. Please revise the diagram for clarity.
The authors should adhere to the MDPI format/guidelines. In the reference section, DOI of the literature should be included.
Comments on the Quality of English LanguageThe English could be improved for clarity and better expression of the theme.
Author Response
Response to Reviewer 3:
The research article studied the role of Bacillus pumilus (endophyte) and a mycorrhizal complex, and application (individual and coinoculation) on the growth and plant performance of Pisum sativum. The results showed a significant improvement in plant growth, photosynthetic efficiency, and plant productivity both on individual applications and when applied together. In the era of sustainable agriculture, the application of bio-based methods is integral to boosting plant growth and productivity. P. sativum being an important food crop, the article defines a promising theme to explore how microbial biostimulants offer feasible solutions in agriculture and maintain environmental sustainability.
Dear Reviewer,
Thank you for your positive and insightful feedback on our manuscript. We appreciate your recognition of the significance of our study. We are pleased that you found our research relevant to sustainable agriculture and environmental sustainability. We value your suggestions and will address them in the revised manuscript.
Thank you again for your time and constructive feedback.
I have some suggestions for the improvement of the manuscript:
Considering the limitations associated with the large-scale application of biostimulants, although the results are promising, what are the future directions in the study? How can the existing challenges be overcome for the field application of the microbial consortia (B. pumilus and mycorrhizal complex)? Discuss.
Thank you for your comment. A large-scale trial of the current experiments will follow to test the ability of the fungal and bacterial inocula to perform under field conditions. This study will focus on the establishment of microbial interactions with Pisum sativum and assess their effects on yield improvement. Addressing challenges such as scalability, cost-effectiveness, and environmental variability will be key in overcoming the limitations associated with field application.
What is the author’s perspective on taking the research study and developing a biostimulant based on microbial consortia for the growth of not just pea plants but also other economically important crops?
Thank you for your comment. The extension of the inoculation technique to other economically important crops is planned as part of our ongoing research. We also aim to evaluate its effectiveness under field conditions, particularly in addressing challenges such as drought and salt stress, which are critical for sustainable agriculture.
In the introduction section, the key biostimulants developed and commercially marketed to improve plant growth and productivity may be discussed. Is there any research on the application of microbial biostimulants in pea plants? What were the microbial consortia and the key results?
Thank you for your valuable feedback. We have updated the introduction to discuss key biostimulants for improving plant growth. Recent research, such as the two-year field study by Ilahi et al. (2025), showed that inoculating Pisum sativum with plant growth-promoting bacteria, including Rhizobium laguerreae and Erwinia sp. strains, significantly improved productivity and biometric parameters. The study also revealed increased bacterial and AM fungal diversity, emphasizing the synergistic role of AMF-bacterial interactions in enhancing soil health and pea growth. These findings are now cited in the revised manuscript.
Please be consistent with the use of scientific names; full names at the first mention, then abbreviated forms may be used. Line 122- maize plant….then Z. mays.
Line 127….seeds of maize (Zea mays L.). Similarly, for Bacillus pumilus.
Thank you for the helpful guidance. We have ensured consistency in the use of scientific names throughout the manuscript, using full names at the first mention and abbreviated forms thereafter
Line 151-156: What was the protocol followed for the production of bacterial inoculum? Please cite the reference study.
Thank you for your comment. The bacterial strain was purified through several centrifugation steps, and the absorbance was checked to ensure a suspension of approximately 2 × 10⁸ CFU/mL. The protocol followed for the production of the bacterial inoculum is based on Slimani et al. (2022), which we have now cited in the revised manuscript.
Figure 1. Phylogenetic tree of bacterial strains. The figure needs to be regenerated and clear, the font size must be increased, and evolutionary distances should be mentioned. The software used for phylogenetic tree construction must be reported in the diagram for clarity.
We thank the reviewer for these constructive suggestions. As requested, we have regenerated the phylogenetic tree to improve clarity. The updated figure now includes readable font sizes, branch length distances, and bootstrap values to reflect evolutionary relationships. Furthermore, we now specify the methods used in the figure legend: 16S rRNA sequences were aligned using Clustal Omega, and the tree was constructed using the Neighbor-Joining method. The resulting tree was visualized and edited using iTOL (Interactive Tree of Life). These improvements are reflected in the revised Figure 1 and its updated legend.
In the biostimulant treatments study, the results suggested that application of AMF on P. sativum plant improved plant height, collar diameter, leaf production, and pod and flower production was better than the endophyte B. pumilus application. Although the study focuses on B. pumilus, AMF consortia can be further developed and optimized as a biostimulant.
Thank you for your insightful comment. Indeed, the combination of AMF and B. pumilus demonstrated the most pronounced improvements, showing a synergistic effect across all measured parameters. We agree that further development and optimization of AMF consortia, particularly in combination with B. pumilus, could offer promising biostimulant applications. Field validation studies are essential to confirm these findings under real-world conditions and to assess the long-term efficacy of B. pumilus both individually and in conjunction with AMF.
In Figure 6. Multivariate analysis of morphological, biomass, and chlorophyll……9A) Pearson correlation heatmap. Some of the parameters are overlapping and are not clear. Please revise the diagram for clarity.
Thank you for the comment. Figure 6 has been revised to improve the clarity of all displayed parameters.
The authors should adhere to the MDPI format/guidelines. In the reference section, DOI of the literature should be included.
Thank you for pointing this out. Suggestions are applied and highlighted.
The English could be improved for clarity and better expression of the theme.
Thank you for the comment. We have revised the manuscript to improve the clarity and expression of the language.
